# Prediction of Pharmacokinetics for CYP3A4-Metabolized Drugs in Pediatrics and Geriatrics Using Dynamic Age-Dependent Physiologically Based Pharmacokinetic Models

**DOI:** 10.3390/pharmaceutics17020214

**Published:** 2025-02-07

**Authors:** Jing Han, Zexin Zhang, Xiaodong Liu, Hanyu Yang, Li Liu

**Affiliations:** Center of Drug Metabolism and Pharmacokinetics, School of pharmacy, China Pharmaceutical University, Nanjing 210009, China; 3222010272@stu.cpu.edu.cn (J.H.); 1821010211@stu.cpu.edu.cn (Z.Z.); xdliu@cpu.edu.cn (X.L.)

**Keywords:** CYP3A4, dosing regimen, physiologically based pharmacokinetic model, pediatrics, geriatrics

## Abstract

**Background/Objectives**: The use of medicines in pediatrics and geriatrics is widespread. However, information on pharmacokinetics of therapeutic drugs mainly comes from healthy adults, and the pharmacokinetic parameters of therapeutic drugs in other age stages, including pediatrics and geriatrics, are limited. The aim of the study was to develop a dynamic age-dependent physiologically based pharmacokinetic (PBPK) model to predict the pharmacokinetics of drugs in humans at different ages. **Method**: The PBPK models characterizing dynamic age-dependence were developed in adults (20–59 years old) and 1000 virtual individuals were constructed. Four CYP3A substrates, namely midazolam, fentanyl, alfentanil and sufentanil, served as model drugs. Following validation using clinic observations in adult populations, the developed PBPK models were extrapolated to other age populations, such as pediatrics and geriatrics, via replacing their physiological parameters and pharmacokinetic parameters, such as organ volume, organ blood flow, clearance, f_u,b_ and K_t:p_. The simulations were compared with clinic observations in corresponding age populations. Midazolam served as an example, the dose transitions between adult pediatrics and adult geriatrics were visualized using the developed PBPK models. **Results**: Most of observed plasma concentrations fell within the 5th–95th percentile of the predicted values in the 1000 virtual individuals, and the predicted AUC_0–t_ and C_max_ were almost within between 0.5 and 2 times of the observations. The optimization of dosages in pediatrics and geriatrics were further documented. **Conclusions**: The developed PBPK model may be successfully used to predict the pharmacokinetics of CYP3A4-metabolized drugs in different age groups and to optimize their dosage regiments in pediatrics and geriatrics.

## 1. Introduction

There has been a pressing need to enhance the safety of medications for pediatric patients and elderly patients. The number of pediatric clinical studies is substantially lower than the number of adult clinical studies due to ethical and clinical trial acceptance, leading to pediatric drug development lagging behind adult drug development [1]. The dosage regiment of drugs used for children is often based on scaled-down adults. However, the organs/tissue and physiological functions of children are always altered along with their growth and development. These alterations in turn affect the pharmacokinetic behaviors of drugs and their clinic efficiencies, leading to differences between pediatrics and adults, even among children in different age stages [2]. According to the FDA and the International Conference on Harmonization (ICH), the categorizations of the pediatric population are premature (gestational age ≤ 36 weeks), newborns (birth to 1 month of age), infants (1–23 months of age), children (2–11 years), and adolescents (12–16/18 years) [3].

The global elderly population is rapidly growing. It was estimated that 16% of the global population will be older than 65 years by 2050 [4]. Elderly peoples are divided into three categories (‘elderly adults’ (61–75 years), ‘old’ persons (76–90 years), and the ‘oldest old’ (over 90 years)) [5] or four categories (the late middle age (60–64 years), the young-old (65–74 years), old-old (75–84 years) and oldest old (85 years and older)) [6]. Most of these elder people often suffer from some diseases, consuming an average of two to five medications per day [7]. However, the pharmacokinetic information of the drugs used in elderly populations is often limited, and elderly patients often receive the same dosage regimen of drugs used for younger patients [8]. Accumulating evidence has demonstrated that some physiological functions of elderly people can be altered along with increasing age [9,10,11], including decreases in gastrointestinal motility, gastrointestinal blood flow, lean body mass, serum albumin, hepatic size, hepatic blood flow, renal blood flow and glomerular filtration rate, as well as increases in gastric pH and adipose mass [9]. For example, the liver volume and liver blood flow of 91-year-old men were, respectively, decreased to be 70% and 57% of that 21-year-old men [12]. Creatinine clearance (CL_CR_) declined linearly across the age span [10,13]. It was reported that CL_CR_ decreased from 140 mL/min/1.73 m^2^ at an age of 30 years to 97 mL/min/1.73 m^2^ at 80 years of age [13]. The alterations in physiological functions of elderly people may affect the pharmacokinetics of drugs used in elderly patients, in turn altering the clinical efficiencies of the drugs. A clinic trial showed that the area under the curve (AUC) of triazolam following oral 0.25 mg triazolam increased and clearance decreased with age (20- to 75-year-old men). The AUC of triazolam increased from 9.8 ± 1.4 ng.h/mL in the young group (20–36 years) to 17.2 ± 4.0 ng.h/mL in the elderly group (60–75 years), accompanied by the greater pharmacodynamics effects of triazolam [14]. The AUC values of S-verapamil and R-verapamil in elderly men (68 ± 5 years) following oral 80 mg verapamil were, respectively, 4.3-fold and 2.7-fold of younger peoples (21 ± 2 years). Consistently, the mean pulse rate in the elderly group was significantly decreased (at 1.5 and 4 h following verapamil) and the mean PQ interval was significantly prolonged (at 1.5 h following verapamil), which did not occur in the young group [15].

The physiologically based pharmacokinetic (PBPK) model is established on the basis of the physiological and biochemical properties of organisms, the anatomical properties of organisms and the thermodynamic properties of drugs. The PBPK model can be used to assess the effects of physiological and pathophysiological changes on the pharmacokinetics of drugs and has been used to predict alterations in the pharmacokinetics of drugs used in pediatric [1,2,16,17] and elderly patients [8].

Over 70% of clinical drugs are metabolized by cytochrome P450 enzymes. CYP3A4/5 represents the most prevalent form of hepatic CYP450 enzymes, accounting for approximately 30–60% of the total hepatic CYP450 and mediating a metabolism of about 50% clinical drugs [18,19,20]. Several reports have demonstrated that the age significantly affects CYP3A4 activity. It was reported that the total intrinsic clearance of triazolam at hepatic microsomes from men aged at 14–20 years, 21–40 years, 41–60 years and 61–72 years were 25.2, 89.8, 78 and 20.6 μL/min/mg protein. Consistently, the levels of hepatic CYP3A protein in hepatic microsomes were, respectively, 17.1, 425.6, 127.3 and 43.7 pmol/mg protein [21]. The activity of hepatic CYP3A, indexed as a clearance of midazolam in preterm neonates, was also less than that in term neonates [22].

Midazolam is a short-acting benzodiazepine with anxiolytic, muscle-relaxant, anticonvulsant, sedative, hypnotic and amnesic properties. Midazolam is widely used for sedation and pre-surgical medication. Midazolam is also one of the most widely used sedatives in the “neonatal intensive care unit” [23]. Alfentanil, fentanyl and sufentanil are synthetic opioid analgesics which act as specific opioid receptors. They are widely used as analgesics to supplement general anaesthesia for various surgical procedures over the entire human age range [24]. Currently, fentanyl is the most frequently used opioid analgesic in neonatal intensive care units [25]. Fentanyl, alfentanil and sufentanil are mainly metabolized by hepatic CYP3A4 enzyme [26,27,28].

The aim of this study was to develop a dynamic age-dependent PBPK model to predict drug pharmacokinetics over the entire human age range from preterm neonates to the oldest old people using the CYP3A4-mediated metabolized drugs midazolam, fentanyl, alfentanil and sufentanil as model drugs and to validate the developed dynamic age-dependent PBPK model using clinical observations. A sensitivity analysis was performed to assess the impact of variability in physiological and biochemical parameters on the pharmacokinetics of the tested drugs. The developed dynamic age-dependent PBPK model was tried to facilitate dose optimization in pediatric and elderly patients.

## 2. Method

### 2.1. General Workflow

The workflow for developing a dynamic age-dependent PBPK model was shown in Figure 1. A dynamic age-dependent PBPK model was developed to predict pharmacokinetics of drugs metabolized by CYP3A over the entire human age range from preterm neonates to the oldest old people. One thousand virtual individuals were constructed. In order to simulate the inter-individual and intra-individual variability of these parameters, the exponential model and additive residual error model were used. The model used the first-order conditional Lindstrom–Bates (FOCE L-B) estimation method. The effective permeability coefficient (P_eff_) [29,30] of drugs across the intestinal wall, free fraction (f_u,b_) [31,32,33,34] of drug in blood, hepatic intrinsic clearance (CL_int,l_) [35], intestinal intrinsic clearance (CL_int,i_) [35], the ratio of drug concentration in blood to plasma (R_b_) [30,31] and the total amount of hepatic microsomal protein (PBSF) [35] were set as the random effect parameters of the virtual individuals, and these parameters were assumed to vary between 0.67 and 1.5 times of the ideal values at the corresponding age stages, based on the variation of clinical and non-clinical observations. Following validation using clinical observations in adults from literature, the developed PBPK model was extrapolated to all different age stages, including pediatrics and geriatrics, by replacing the values of system-specific model parameters (anatomical, biochemical and physiological parameters). The pharmacokinetics of the tested drugs were predicted in 1000 virtual individuals of corresponding age stages and compared with clinical observed pharmacokinetic data from the literature. The validated PBPK model was then used to optimize the drug dosage for populations of the corresponding age stages (pediatrics and geriatrics) based on drug plasma exposure in adults.

### 2.2. Development of Dynamic Age-Dependent PBPK Model

A dynamic age-dependent PBPK model (Appendix A) was developed to describe the pharmacokinetic profiles of midazolam, fentanyl, alfentanil and sufentanil in the plasma of the entire human age range following oral or intravenous administrations. The dynamic age-dependent PBPK model consists of stomach, intestine, liver, spleen, kidneys, muscle, adipose tissue, skin, heart, brain, lungs, and other tissue, which are connected by the blood circulatory system. The intestine is divided into the duodenum, jejunum, ileum, cecum and colon according to the characteristics of physiology and anatomy. Each intestinal segment is further divided into the intestinal wall and lumen. Drugs are mainly eliminated via the liver and kidneys. Drug metabolism also occurs at the wall of duodenum, jejunum and ileum. For oral administration, drug absorption only occurs in duodenum, jejunum and ileum. The essential structure of the dynamic age-dependent PBPK model (Appendix A) and its corresponding mass equations are illustrated in the Appendix A.

### 2.3. Development and Optimization of PBPK Models in Adults

The drug-specific parameters and physiological parameters of adults used in the simulations were, respectively, listed in Appendix A. Plasma concentrations of the four drugs in 1000 virtual individuals were predicted on Phoenix WinNonlin software (Version 8.4, Certara, Radnor, PA, USA). The simulation results were further compared with clinical observations.

The plasma concentration profiles of drugs in 1000 virtual adult individuals were simulated using the PBPK model. The 5th, 50th and 95th percentiles of plasma concentration profiles were obtained and compared with clinical observations. The pharmacokinetic parameters, such as the concentration–time curve (AUC_0–t_) and peak concentration (C_max_) were also estimated from the 50th percentiles of plasma concentration profiles.

Two aspects were used to evaluate the successful development of the dynamic age-dependent PBPK model. (i) The observed plasma concentration profiles fell within the 5th–95th percentiles of simulations based on 1000 virtual individuals; (ii) The simulations using the developed PBPK model were within the 0.5–2.0-fold of clinic observations. In addition, the following two model prediction performance indicators were calculated for simulations (AUC_0–t_ and C_max_). We also evaluate simulations using the average fold error (AFE) and the average absolute prediction error (PE%).

The AFE was defined by the equation [36](1)$$AFE=10^{\frac{1}{n}\sum log\frac{Predi}{Obsi}}$$

The prediction may be considered satisfactory if the AFE is between 0.8 and 1.25, passable if the AFE is within 0.5–0.8 or 1.25–2 and poor if the AFE is within 0–0.5 or above 2.

The average absolute prediction error (PE%) is defined by equation [36](2)$$PE(\%)=Geomean\left(\left|\frac{Prei-Obsi}{Obsi}\right|\right)\times 100$$

The prediction may be considered satisfactory if the PE is less than 25%, passable if the PE is between 25 and 50% and poor if the PE is ≥50%.

### 2.4. Extrapolation to the Pediatric PBPK Model

The PBPK model, following validation and optimization in adults, was extrapolated to pediatrics. One thousand virtual pediatric individuals (newborns, infants, children and adolescents) were constructed. Physiological parameters (such as organ volumes and organ blood flow) and parameters related to drugs at corresponding age stages were adjusted as follows. Physiological parameters (such as organ volumes and organ blood flow) of pediatrics along with age were estimated using the equations listed in Appendix A.

The body weight (BW, kg) and height (H, cm) of pediatrics were estimated using equation [37].(3)$$Y=\frac{a+b\times A}{1+c\times A+d\times A^{2}}$$
where Y is body weight or height. A is age (year). The correlation coefficients a, b, c and d are shown in Appendix A.

The gastric emptying time of neonates (0–27 days), infants (1–23 months), children (2–11 years), adolescents (12–18 years) and adults (>18 years) were, respectively, 68 min (54–82 min), 41 min (12–70 min), 41 min (12–70 min), 75 min (12–138 min) and 62.5 min (5–120 min); no obvious age-dependent differences were observed. The intestinal transit time in pediatrics was also similar to that in adults [38]. Here, the gastric emptying time and intestinal transit time in pediatrics were assumed to be similar to those for adults.

The intestinal radius (r, m) was associated with body surface area (BSA), which may be calculated using the following equation [39]:r = (0.016 × BSA + 0.0159)/2(4)
where BSA may be estimated using the following equations [40,41]:BSA = 0.007184 × BW^0.425^ × H^0.725^, BW > 15 kg(5)
andBSA = 0.024265 × BW^0.5378^ × H^0.3964^, BW < 15 kg(6)

The activity of the hepatic CYP3A4 increased along with growth and development, which may be illustrated by equation [39]. The ratio (R_CYP3A,pediatric_) of hepatic CYP3A activity as follows:(7)$$R_{CYP3A,pediatric}=\frac{A^{0.83}}{0.31+A^{0.83}}$$

The impact of gestational age on the development of newborns and preterm infants was of critical importance. The activity of the hepatic CYP3A4 in neonates and preterm infants was illustrated by an equation as follows [42]:R_CYP3A,neonate/preterm_ = 0.003 + 0.1331 × PMA(8)
where R_CYP3A,neonate/preterm_ is the ratio of hepatic CYP3A activity in neonates or preterm infants to adults. The PMA is the postmenstrual age (year). PMA is the sum of the gestational age (GA) and postnatal age (PNA).

Hepatic microsome protein also varied with age, and the microsomal protein per gram of liver (MPPGL, mg protein/g liver) was estimated using equation [43].(9)$${\mathrm{MPPGL}}_{\mathrm{pediatric}}=10^{\left(1.407+0.0158\times A-0.00038\times A^{2}+0.0000024\times A^{3}\right)}$$

The intrinsic hepatic clearance (CL_int,pediatric_) in pediatrics was illustrated by the following equation:CL_int,pediatric_ = CL_int,adult,l_ × R_CYP3A,pediatric_ × MPPGL_pediatric_ × LW_pediatric_(10)
where CL_int,adult,l_ and LW_pediatric_ are the intrinsic hepatic clearance of the drugs in the hepatic microsomes (mL/min/mg) of adults and the liver weight of pediatrics, respectively.

The expressions of intestinal CYP3A were altered along with age; the fraction (R_CYP3A,pediatric,i_) of adult intestinal CYP3A may be estimated using the following equation:(11)$$R_{\mathrm{CYP}3A,\mathrm{pediatric},i}=\frac{0.639\times A}{2.36+A}+0.42$$

The intrinsic intestinal clearance (CL_int,pediatric,i_) in pediatrics was illustrated by the following equation:CL_int,pediatric,i_ = CL_int,pediatric,l_ × R_CYP3A,pediatric,i_ × A_adult,i_(12)
where CL_int,pediatric,l_ and A_adult,i_ are the hepatic clearance of the drugs in the CYP3A (mL/min/pmol) of pediatrics and the CYP3A4 abundance in adults’ intestines, respectively.

The levels of plasma albumin (ALB) and alpha1-acid glycoprotein (AAG) levels also increased along with age, leading to a decrease in f_u,b_. The levels of plasma ALB (g/L) in preterm infants and pediatrics were, respectively, estimated using the following equations.

For preterm infants [42](13)$${\mathrm{ALB}}_{\mathrm{preterm}}=\frac{41.3\times {PMA}^{2.70}}{0.383^{2.70}+{PMA}^{2.70}}$$

For pediatrics [39,44],ALB_pediatrics_ = 1.1287 × lnA + 33.746(14)

Levels of plasma AAG in pediatrics were calculated using the following equation.AAG_pediatrics_ = AAG_adult_ × R_AAG,pediatrics_(15)
where R_AAG,pediatrics_ and AAG_adult_ are the ratios of the AGG level in pediatrics and the AGG level in adults, respectively. Here, the AGG level in adults was set to be 0.61 g/L [45]. The R_AGG,pediatrics_ was calculated using following equation [46]:R_AAG,pediatrics_ = 0.01137 × days + 53.4(16)
where the “days” represent the number of days after birth.

The unbound fraction (f_u,pediatrics_) of drugs in the plasma of pediatrics was estimated using the following equation [46]:(17)$$f_{u,\mathrm{pediatrics}}=\frac{1}{1+\frac{P_{pediatrics}\times (1-f_{u,adult})}{P_{adult}\times f_{u,adult}}}$$
where f_u,adult_ is the unbound fraction of drug in plasma of adults. P_pediatrics_ and P_adult_ are the levels of plasma ALB (or AAG) in pediatrics and adults, respectively.

Following obtaining physiological parameters and pharmacokinetic parameters in pediatrics, 1000 virtual pediatric individuals were constructed. The pharmacokinetic profiles of drugs in the 1000 virtual pediatric individuals were simulated. The 5th, 50th and 95th percentiles of plasma concentration-profiles were obtained and compared with clinical observations from pediatrics at corresponding age stages.

### 2.5. Extrapolation to the Geriatrics PBPK Model

The physiological parameters including height, weight, body surface area, tissue and organ volume, and blood flow in the elderly were altered along with increasing ages, which were estimated using equations listed in Appendix A.

In consideration of the hepatic metabolism of drugs, it was reported that, in predicted clearances from 30 to 35 years, the MPPGL decreased along with increasing adult age until approximately 80–85 years. Then, the MPPGL failed to decrease further with advancing age. More importantly, the MPPGL at 85–90 years and 90–95 years were higher than those at 80–85 years [47]. The relationships between the MPPGL in the elderly and age were illustrated using the following equations:MPPGL_elderly_ = 0.0001653 × A^3^ − 0.02739 × A^2^ + 1.143 × A + 25.52, A < 80 years(18)
andMPPGL_elderly_ = −0.08155 × A^2^ + 14.48 × A − 612.7, A > 80 years(19)
where A is age (year).

It was reported that the activity of the CYP3A4 enzyme in the elderly population decreased by 8% per decade [48,49]; thus, the ratio of CYP3A in the elderly to 40-year-old adults was illustrated by following equation(20)$$R_{CYP3A,elderly}={\left(1–8\%\right)}^{(A-40)/10}$$

The intrinsic hepatic clearance of drug in the elderly (CL_int,elderly_) can be expressed as follows:CL_int,elderly_ = CL_int,adult,l_ × R_CYP3A,elderly_ × MPPGL_elderly_ × LW_elderly_(21)
where CL_int,adult,l_ is the intrinsic hepatic clearance of the drug in the microsomes (mL/min/mg) of adults; the MPPGL_elderly_ and LW_elderly_ are the productivity of hepatic microsomes and liver weight in the elderly, respectively.

The intrinsic intestinal clearance of drugs in the elderly (CL_int,elderly,i_) was assumed to be the same as in adults, which may be corrected by the body weight at a special age.

It was shown that, from 20 to 90 years, ALB concentration (g/L) decreased with increasing age [47], which was illustrated by the following equation:ALB_elderly_ = 51.4 − 0.107 × A(22)

However, the AAG concentration did not change with age. Here, the AAG level in the elderly was set to be 0.58 g/L.

The unbound fraction (f_u,elderly_) of drugs in plasma in the elderly was estimated using Equation (15).

Following the obtaining of physiological parameters and pharmacokinetic parameters in the elderly, 1000 virtual individuals of different elderly age stages were constructed. The pharmacokinetic profiles of drugs in the 1000 virtual elderly individuals were simulated. The 5th, 50th and 95th percentiles of plasma concentration profiles were obtained and compared with clinical observations at corresponding elderly age stages.

### 2.6. Sensitivity Analysis

Parameters such as physiological parameters (Q_l_, PBSF and K_ti_) and pharmacokinetic parameters (f_u,b_, P_eff_ and CL_int,l_) were selected for sensitivity analyses using midazolam as a model drug. In local sensitivity analysis evaluations, the variabilities of Q_l_ [50,51], f_u,b_ [31,32] and K_ti_ [52] was set at 0.5, 1 and 2 times, while the variabilities of PBSF and CL_int,l_ were set at 0.1, 1 and 10 times [35]. Midazolam was given to adults via an oral dose (10 mg) [53] or an intravenous dose of 0.75 mg over 2 min [54]. A global sensitivity analysis (GSA) evaluation was also performed for the identification of the more accurate parameter’s contribution. The variability of parameters was set as the same as the local sensitivity analysis. A GSA was performed with the R (Version 4.4.2), RStudio (Version 2024.12.0+467) and RXODE2 package. Total order sensitivity indices and first-order sensitivity indices was estimated using the SOBOL method [55] for the GSA. Corresponding script are included in the Appendix A.

### 2.7. Pediatrics and Geriatrics Dose Optimization

Midazolam served as an example of clinical dose optimization in pediatrics and geriatrics. According to age, the patients were divided into 0–0.083 year old, 0.083–0.5 year old, 0.5–1 year old, 1–2 years old, 2–5 years old, 5–9 years old, 9–12 years old, 12–15 years old, 15–18 years old, 19–59 years old, 60–65 years old, 65–75 years old, 75–85 years old and ≥85 years old. The pediatric and the geriatric dosages of midazolam were optimized to achieve AUC and C_max_ values of midazolam that were similar (within 15% deviation) to those following i.v. 0.75 mg midazolam [54].

### 2.8. Collection of Data

The pharmacokinetic data of the midazolam, fentanyl, alfentanil and sufentanil following oral or intravenous dose to the entire human age range (from preterm to the oldest old) were collected from the data published on PubMed based on the following criteria. (1) The tested drugs must be mainly metabolized by hepatic CYP3A. The fraction of the CYP3A-mediated metabolism was over 70%. (2) The pharmacokinetics of the tested drugs were examined in the pediatric patients, adults and the elderly, whose plasma concentrations profiles or pharmacokinetic parameters were simultaneously shown. The plasma concentration profiles may be digitized using the Engauge Digitizer. (3) The tested drugs were administrated via an oral or intravenous route. (4) The clinical pharmacokinetic data might come from different reports or patients, but the diseases did not affect the pharmacokinetics of the tested drugs.

## 3. Result

A total of 166 sets of data from the clinical studies involving four CYP3A4 substrate drugs were collected (Appendix A), including 85 sets for midazolam, 54 sets for fentanyl, 15 sets for alfentanil and 12 sets for sufentanil; 43 sets of data were for pediatrics (23 sets for midazolam, 12 sets for fentanyl, 5 sets for alfentanil and 3 sets for sufentanil); 102 sets of data were for adults (52 sets for midazolam, 36 sets for fentanyl, 9 sets for alfentanil and 5 sets for sufentanil); 21 sets of data were for the elderly (10 sets for midazolam, 6 sets for fentanyl, 1 set for alfentanil and 4 sets for sufentanil). Forty-one sets of data were for pediatrics (21 sets for midazolam, 12 sets for fentanyl, 5 sets for alfentanil and 3 sets for sufentanil).

### 3.1. Model Validation in Adults

The established PBPK model was employed to forecast the pharmacokinetic profiles of the four drugs in 1000 virtual adults, with the results being subsequently compared with the clinical observations (Figure 2, Figure 3, Appendix A). The results demonstrated that almost all of the observed plasma concentrations fell within the 5th–95th interval of the simulations from 1000 virtual adults. Furthermore, the pharmacokinetic parameters such as AUC and C_max_ were estimated using the 50th percentile of simulated plasma concentrations and compared with the clinical observations (Appendix A). The results demonstrated that the almost predicted AUC (97/97) and C_max_ (28/28) were within a range of 0.5–2-folds of the observed values and that the AFE (0.9) and PE (15.1%) were satisfactory, indicating the successful development of a PBPK model for the four drugs in adults.

### 3.2. PBPK Model in Pediatrics

The developed PBPK models were extrapolated to pediatrics to predict the pharmacokinetic profiles of the four drugs following administration to newborns, infants, children and adolescents. The 5th, 50th and 95th percentiles of plasma concentration profiles from 1000 virtual pediatrics were obtained and compared with clinical observations from pediatrics at corresponding age stages (Figure 4 and Appendix A). The results showed that, among the selected 43 sets of data, except for two sets of data, other clinic observations almost fell within the 5th to 95th percentile range of the simulations. The pharmacokinetic parameters were estimated from the 50th percentile of plasma concentration profiles and compared with clinical observations (Appendix A) The results demonstrated that the predicted AUC (34/35) and C_max_ (5/6) were within a range of 0.5–2-folds of the observed values, and that the AFE (1.1), PE (24.0%) was also satisfactory. These results indicated successful modelling of four drugs in pediatrics.

### 3.3. PBPK Model in Geriatrics

The developed PBPK models were further extrapolated to geriatrics to predict the pharmacokinetic profiles of the four drugs following administration to the late middle age, the young-old, old-old and oldest old. The 5th, 50th and 95th percentiles of plasma concentration profiles from 1000 virtual pediatrics were obtained and compared with clinical observations from geriatrics at corresponding age stages (Figure 5). The pharmacokinetic parameters were estimated from 50th percentile of plasma concentration-profiles and compared with clinical observations (Appendix A). The results demonstrated that almost all of the observed plasma concentrations fell within the 5th–95th interval of the simulations from 1000 virtual adults. In addition, the results also demonstrated that the predicted AUC (19/19) and C_max_ (4/5) were within a range of 0.5–2-folds of the observed values and that the AFE (1.1), PE (24.2%) was also satisfactory. The above results indicated successful modelling of four drugs in geriatrics.

### 3.4. Sensitivity Analysis of Model Parameters

The physiological parameters Q_l_, PBSF, K_ti_ and the pharmacological parameters f_u,b_, CL_int,l_, P_eff_ were selected for local sensitivity analyses to observe the effects of these parameters on the plasma concentrations of midazolam following oral administration and intravenous administration of midazolam, fentanyl, alfentanil and sufentanil (Figure 6A–F). The findings indicated that alterations in PBSF and CL_int,l_ exerted a more pronounced influence on the plasma concentrations of the four drugs administered intravenously followed by f_u,b_ and Q_l_. These results align with our hypothesis that these four drugs are primarily metabolized by hepatic CYP3A4 and, thus, an increase in PBSF and CL_int,l_ would consequently impact the metabolic rate and the plasma concentrations. Furthermore, following oral administration of midazolam, an increase in P_eff_ will elevate C_max_, while K_ti_ exerted a diminished influence on the midazolam concentrations. A global sensitivity analysis revels the SOBOL sensitivity indices of the six parameters in the PBPK model of midazolam (Figure 6G). Compared with other parameters, PBSF, CL_int,l_ and f_u,b_ are the key parameters of the model output (AUC_0–24_). When the first-order sensitivity indices (S1) are considered, the three parameters were not as critical as when the total order sensitivity indices (ST) were considered. This suggested that there is a strong interaction between PBSF, CL_int,l_ and f_u,b_. In comparison, the effect of Q_l_ is weaker and the effects of K_ti_ and P_eff_ are much smaller, based on ST or S1.

### 3.5. Virtual Simulation for Pediatric and Geriatrics Dose Optimization

The effects of age on the AUC and C_max_ of midazolam at clinically recommended doses were also recorded (Figure 7). The recommended clinical dose of midazolam is 0.05 mg/kg for <6 months, 0.05–0.01 mg/kg for pediatrics aged 6 months–6 years, 0.025–0.05 mg/kg for pediatrics aged 6–12 years, 0.5–1 mg for pediatrics aged > 12 years (as well as adults) and 1–1.5 mg for geriatrics [53]. The results showed that the AUC and C_max_ of midazolam were higher than those of adults at the recommended clinical dose in the pediatric groups, with special attention being paid to the fact that the AUC and C_max_ values of 0.17 mg midazolam injected into infants at 1 month (≈0.083 year) of age were 6.16 and 2.47 times higher than those of adults injected with recommended 0.75 mg midazolam. In geriatrics, the AUC and C_max_ were within a range of 1.5–2.5 times higher than those of adults at the recommended clinical dose.

Direct comparisons were made by a Box–Whisker analysis to account for the pharmacokinetic variability in pediatrics and the elderly. We constructed nine groups of pediatrics and four groups of geriatrics for the dose design of midazolam. As shown (Figure 7), the dose of sedated midazolam should be adjusted to 8.4 μg/kg for 0–0.083 year, 11.91 μg/kg for 0.083–0.5 year, 16.32 μg/kg for 0.5–1 year, 17.59 μg/kg for 1–2 years, 17.70 μg/kg for 2–5 years, 15.43 μg/kg for 5–9 years, 13.37 μg/kg for 9–12 years, 12.88 μg/kg for 12–15 years, 12.40 μg/kg for 15–18 years, 8.3 μg/kg for 60–65 years and 7.82 μg/kg for 65–75 years, 7.38 μg/kg for 75–85 years and 7.24 μg/kg for >85 years of age to make the AUC_0–t_ comparable to that of an adult dose of 0.75 mg of midazolam administered intravenously. To make the C_max_ comparable to that of an adult dose of 0.75 mg of midazolam administered intravenously, the dose of sedated midazolam should be adjusted to 22.41 μg/kg for 0–0.083 years, 18.58 μg/kg for 0.083–0.5 year, 17.10 μg/kg for 0.5–1 year, 18.16 μg/kg for 1–2 years, 19.37 μg/kg for 2–5 years, 17.44 μg/kg for 5–9 years, 14.31 μg/kg for 9–12 years, 13.24 μg/kg for 12–15 years, 12.41 μg/kg for 15–18 years, 8.96 μg/kg for 60–65 years, and 8.82 μg/kg for 65–75 years, 8.68 μg/kg for 75–85 years and 8.63 μg/kg for >85 years. The results showed that optimized dosages of midazolam in pediatrics (<12 years) and geriatrics (>60 years) were lower than recommended dosages.

## 4. Discussion

It is challenging to recruit patients from vulnerable groups, such as pediatrics and geriatrics, for clinical trials of drugs. This is because these groups require additional safeguards during the research process [115]. Despite the increasing need for medications in pediatrics and the elderly, clinical studies rarely target these age groups [116]. Furthermore, a limited number of clinical observations and time points exist. Dosing in pediatrics as well as in the elderly needs to take into account the changes in physiological and biological parameters with ages. Thus, a PBPK model characterizing age-related changes in physiological and biochemical parameters was developed to simulate the disposition of the drugs in both pediatrics and the elderly; this will provide help for the designment of dosages in both populations. Midazolam, fentanyl, alfentanil and sufentanil served as model drugs. The established PBPK model was first validated by plasma drug concentration–time curves after oral or intravenous administration of model drugs in adults. Then, the developed PBPK model was extrapolated to pediatrics and the elderly. The developed PBPK models could reasonably predict the pharmacokinetics of four model drugs in all age groups. The majority of measured values fell within the 5th to 95th percentile range of the predicted values, and the C_max_ and AUC_0–t_ were consistently within 0.5 to 2.0 times of the clinical observations.

During the modelling process, the four model drugs chosen belong to the CYP3A4 substrates, and the activity of the CYP3A4 enzyme is also an aspect that is considered in addition to anatomical and physiological parameters such as organ volume and organ blood flow. Enzyme activities and organ systems are constantly changed with age, especially for preterm neonates. The high variability in preterm neonates may explain the somewhat poorer simulation compared to other age populations [117]. Three equations were used to estimate the levels of ALB based on their age stages. Furthermore, the exposure of drugs at clinically recommended doses in different age groups was evaluated, including AUC_0–t_ and C_max_. Additionally, dose optimization in pediatrics and the elderly was investigated using established PBPK models to ensure the plasma exposures were comparable to those observed in adults, with midazolam serving as an illustrative example.

For midazolam, pediatrics dosing requires special attention. One study counted 1578 pediatric endoscopies with a median age of 10 years and intravenous midazolam doses of 0–0.39 mg/kg, which were higher than the model-optimized dose of 13.37/14.31 μg/kg for ages 9–12 years in Figure 7. The results of adverse reaction statistics showed a 10% incidence of decreased oxygen saturation, 8% incidence of hypotension, 3% incidence of hypertension, and 5% incidence of vomiting [118]. In addition, Pena et al. counted cases observed in the emergency department of an urban pediatric teaching hospital from 1997 to 1998 and showed that, among pediatric patients receiving fentanyl for sedation and analgesia, case statistics showed that 84% of the pediatric patients who experienced adverse reactions had midazolam i.v doses (0.16 or 0.09 mg/kg) beyond the model-recommended values for the corresponding age group (Figure 7), which may explain some of the adverse reactions in some patients, such as oxygen saturations of less than 90% (nine cases) and vomiting (three cases) [119]. In a study that administrated intravenous midazolam and ketamine for sedation in pediatric patients from 5 months to 17 years, it was observed that half of the children treated with 0.3 mg/kg of midazolam developed oxygen desaturation. It was notable that both the initiating (0.1 mg/kg) and terminating doses (0.4 mg/kg) were considerably higher than ten folds of the model-informed dose. This may help to explain the high incidence of adverse effects, including tachycardia (27.9%), increased secretion (17.6%), agitation (13.6%) and oxygen desaturation (8.4%) [120]. Similarly, a trial on 164 children demonstrated that the incidence of paradoxical reactions was significantly higher in children aged 1–3 years old who were sedated with midazolam at 0.1 mg/kg than in children aged 1–3 years old who were sedated at 0.05 mg/kg, both of which were higher than the incidence in children aged 3–5 years old who were sedated at 0.1 mg/kg [121]. The administered doses are at least two fold of the model-recommended dose of the 1–2 or 2–5-years-old groups, respectively (Figure 7).

Furthermore, the administration of fentanyl and midazolam in the elderly population merits particular attention. The metabolism of the same dose of intravenously delivered fentanyl in the elderly presents a risk of reduced clearance and an elevated half-life and AUC in comparison to younger volunteers [85]. Furthermore, another study demonstrated the probability of increased risk of respiratory depression with age for opioids, including fentanyl [122]. A further randomized, double-blind study indicated a notable reduction in mean blood pressure following the intravenous administration of 0.06 mg/kg midazolam in the elderly, while a comparatively minor reduction was observed in the 0.02 mg/kg dosing group. The decrease in oxygen saturation was significantly greater in the 0.06 mg/kg dosing group of older adults over 60 years compared to the 0.02 mg/kg dosing group [123].

This study represents an attempt to develop a PBPK model characterizing different stages of ages. It can be employed to achieve dose optimization in pediatric and geriatric populations. However, there are some limitations to our study. Firstly, there were limited data from clinical studies, some of which had a wide age range. Here, the age and body weight of the indicated patients were their mean values, which may be somewhat different from the patient’s actual situations. In addition, our model drugs are commonly used for intraoperative sedation and anesthesia, and we did not consider the effects of surgery on in vivo pharmacokinetics in the fitting process. A review of the literature revealed that the performance of certain surgical procedures, such as a cardiopulmonary bypass and extra-corporeal membrane oxygenation, during cardiac surgeries may result in unintended consequences in regard to pharmacokinetic parameters [124,125,126,127]. For example, Kumar et al. investigated the effects of a cardiopulmonary bypass on the plasma concentrations and protein binding of continuously infused alfentanil and found that the half-life of alfentanil was prolonged during the cardiopulmonary bypass and that there was a decrease in the plasma concentrations in vivo, as well as a significant decrease in the albumin and α-acid glycoprotein concentrations, corresponding to an increase in f_u,p_ [127]. Another aspect worth pondering is that, when we performed dose optimization, we used AUC_0–t_ and C_max_ as optimization metrics and did not take into account the pharmacodynamics (PD) metrics, which for some drugs, especially antimicrobials, may need to be integrated with PK-PD to assess the dose–effect relationship so that dose optimization will be more comprehensive and accurate. Although gender may affect the pharmacokinetics of some drugs, several studies [49,71,92,128,129,130,131,132,133,134,135,136,137,138,139,140,141,142,143,144,145] have demonstrated that gender did not obviously affect the pharmacokinetics of the four tested drugs. Thus, the effects of gender on pharmacokinetics were not considered in the simulation.

## 5. Conclusions

The dynamic age-dependent PBPK model may successfully be applied to predict the pharmacokinetics of CYP3A4-mediated metabolized substrate drugs over the entire human age range and to design the dosage of CYP3A4 substrates.

## Figures and Tables

**Figure 1 pharmaceutics-17-00214-f001:**
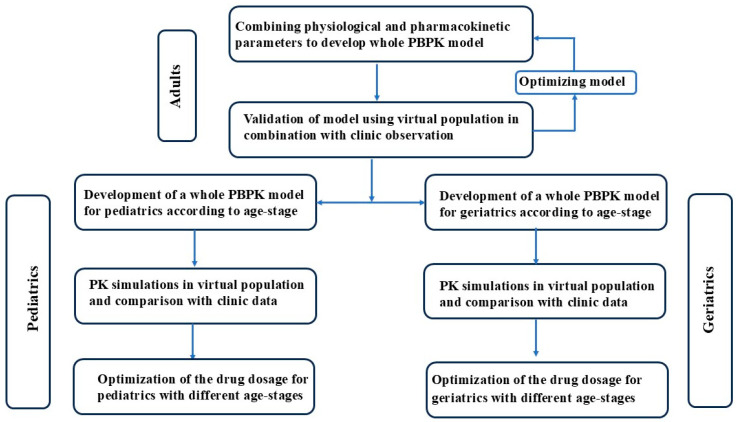
Workflow for developing a dynamic age-dependent PBPK model.

**Figure 2 pharmaceutics-17-00214-f002:**
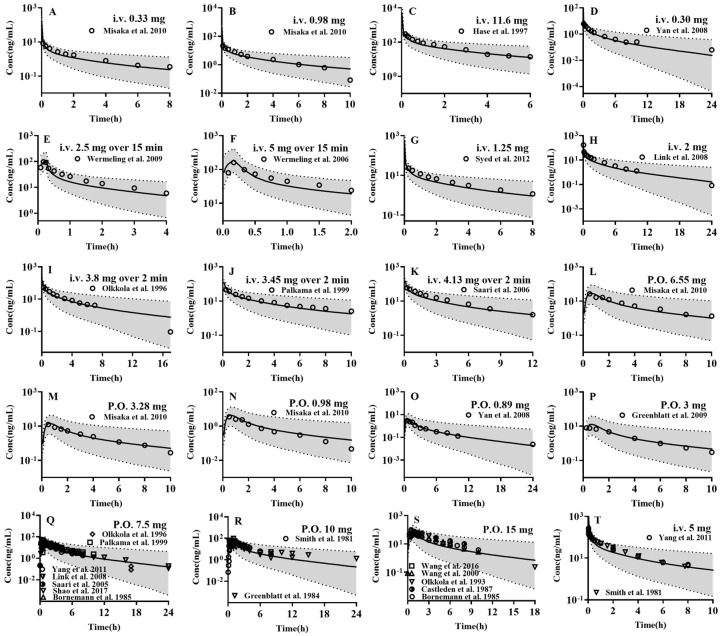
The predicted (lines) and observed (points) plasma concentrations of midazolam following intravenous administration of midazolam (**A**) 5 μg/kg (0.33 mg) in healthy adults [56], (**B**) 15 μg/kg (0.98 mg) in healthy adults [56], (**C**) 0.2 mg/kg (11.6 mg) in orthopaedic surgery adults [57], (**D**) 5 μg/kg (0.30 mg) in healthy adults [58], (**E**) 2.5 mg over 15 min in healthy adults [59], (**F**) 5 mg over 15 min in healthy adults [60], (**G**) 1.25 mg in healthy adults [61], (**H**) 2 mg in healthy adults [62], (**I**) 0.05 mg/kg (3.8 mg) over 2 min in healthy adults [63], (**J**) 0.05 mg/kg (3.45 mg) over 2 min in healthy adults [64], (**K**) 0.05 mg/kg (4.13 mg) over 2 min in healthy adults [65], (**T**) 5 mg in in healthy adults [66,67] or oral administration (**L**) 100 μg/kg (6.55 mg) in healthy adults [56], (**M**) 50 μg/kg (3.28 mg) in healthy adults [56], (**N**) 15 μg/kg (0.98 mg) in healthy adults [56], (**O**) 15 μg/kg (0.89 mg) in healthy adults [58], (**P**) 3 mg in healthy adults [68], (**Q**) 7.5 mg in healthy adults [62,63,64,65,66,69,70], (**R**) 10 mg in healthy adults [67,71], (**S**) 15 mg in healthy adults [29,70,72,73,74]. Solid line, 50th percentile of simulated plasma concentrations; Shadow, 5th–95th interval of the simulated plasma concentrations.

**Figure 3 pharmaceutics-17-00214-f003:**
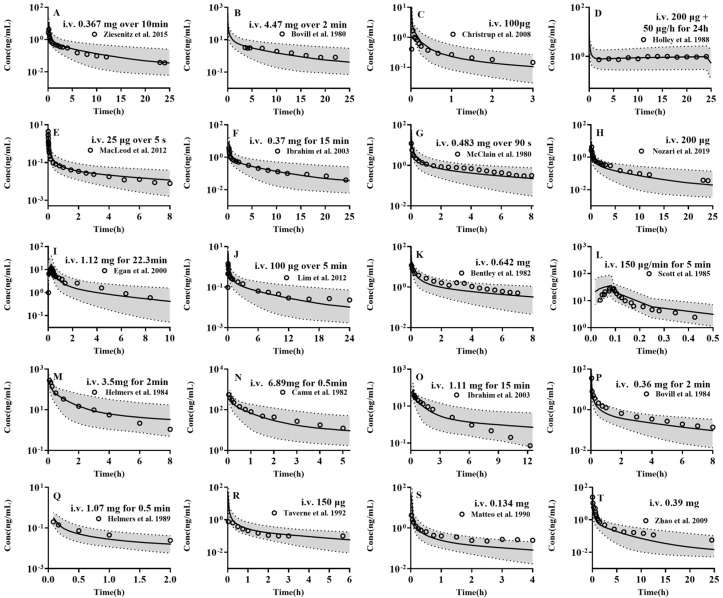
The predicted (lines) and observed (points) plasma concentrations of (**A**–**L**) fentanyl, following intravenous administration of fentanyl (**A**) 5 μg/kg (0.367 mg) over 10 min in healthy adults [75], (**B**) 60 μg/kg (4.47 mg) over 2 min in elective cardiac surgery with cardiopulmonary bypass adults [76], (**C**) 100 μg in surgical removal of both mandibular third molars adults [77], (**D**) loading dose 200 μg followed by 50 μg/h for 24 h in adults undergoing surgery [78], (**E**) 25 μg over 5 s in healthy adults [79], (**F**) 5 μg/kg (0.37 mg) over 15 min in healthy adults [80], (**G**) 6.4 μg/kg (0.483 mg) over 90 s in healthy adults [81], (**H**) 200 μg in craniotomy adults [82], (**I**) 15 μg/kg (1.12 mg) over 22.3 min in healthy adults [83], (**J**) 100 μg over 5 min in healthy adults [84], (**K**) 10 μg/kg (0.642 mg) in adults [85], (**L**) 150 μg/min (0.737 mg) for 5 min in adults undergoing elective surgery involving minimal blood loss [86], (**M**–**O**) alfentanil, following intravenous administration of alfentanil (**M**) 50 μg/kg (3.5 mg) over 2 min in intra-abdominal surgery adults [87], (**N**) 120 μg/kg (6.89 mg) over 0.5 min in healthy adults [88], (**O**) 15 μg/kg (1.11 mg) over 15 min in healthy adults [80], (**P**–**T**) sufentanil, following intravenous administration of sufentanil, (**P**) 5 μg/kg (0.36 mg) over 2 min in adults undergoing surgery [89], (**Q**) 15 μg/kg (1.07 mg) over 0.5 min in adults undergoing surgery [90], (**R**) 150 μg in adults undergoing elective major abdominal surgery [91], (**S**) 2 μg/kg (0.134 mg) in adults undergoing elective neuro surgery [85], (**T**) 7.2 μg/kg (0.39 mg) in adults undergoing surgery [92]. Solid line, 50th percentile of simulated plasma concentrations; Shadow, 5th–95th interval of the simulated plasma concentrations.

**Figure 4 pharmaceutics-17-00214-f004:**
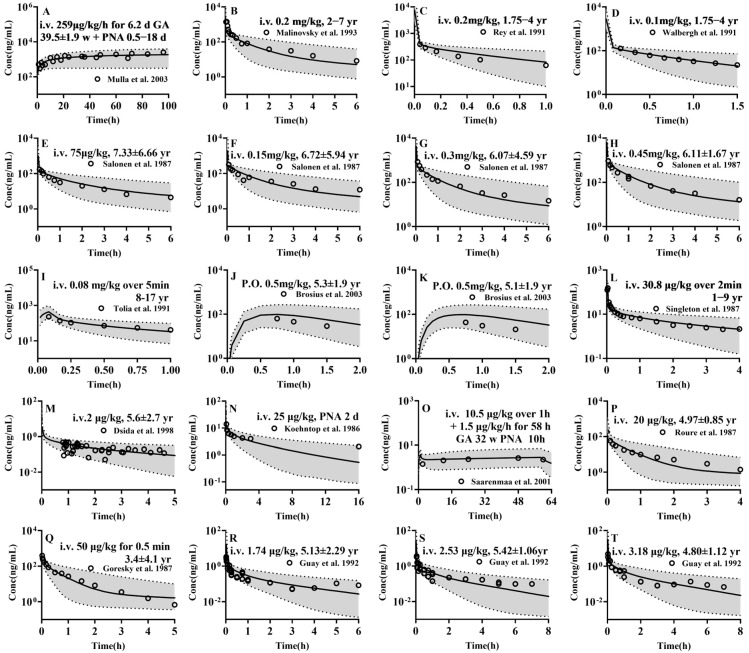
The predicted (lines) and observed (points) plasma concentrations of (**A**–**K**) midazolam, following intravenous administration of midazolam (**A**) 259 μg/kg/h(131.03 mg) for 148.8 h in newborns during ECMO [93], (**B**) 0.2 mg/kg (3.88 mg) in children undergoing minor urological surgery [94], (**C**) 0.2 mg/kg (3.04 mg) in children undergoing minor genito-urinary operations [95], (**D**) 0.1 mg/kg (1.28 mg) in children undergoing elective closure of an asymptomatic atrial septal or ventricular septal defect [96], (**E**) 75 μg/kg (2.46 mg) in children undergoing elective surgery [97], (**F**) 0.15 mg/kg (4.2 mg) in children undergoing elective surgery [97], (**G**) 0.3 mg/kg (6.48 mg) in children undergoing elective surgery [97], (**H**) 0.45 mg/kg (10.035 mg) in children undergoing elective surgery [97], (**I**) 0.08 mg/kg (3.853 mg) over 5 min in children, following oral administration of midazolam [98], (**J**) 0.5 mg/kg (10 mg) in children undergoing surgical procedures with minimal anticipated blood loss [99], (**K**) 0.5 mg/kg (9.55 mg) in children undergoing surgical procedures with minimal anticipated blood loss [99], (**L**–**O**) fentanyl, following intravenous administration of fentanyl, (**L**) 30.8 μg/kg (0.417 mg) in children undergoing nonabdominal surgery [100], (**M**) 2 μg/kg (0.056 mg) in children undergoing elective tonsillectomy [101], (**O**) 10.5μg/kg over 1 h followed by 1.5 μg/kg/h for 58 h in preterm neonates in Intensive Care Unit [102], (**P**,**Q**) alfentanil, following intravenous administration of alfentanil, (**P**) 20 μg/kg (0.355 mg) in children undergoing elective orthopaedic surgery [103], (**Q**) 50 μg/kg (0.762 mg) in children undergoing surgery [104] and (**R**–**T**) sufentanil, following intravenous administration of sufentanil, (**R**) 1.74 μg/kg (32.78 μg) in children undergoing elective anaesthesia and surgery [105], (**S**) 2.53 μg/kg (47.54 μg) in children undergoing elective anaesthesia and surgery [105], (**T**) 3.18 μg/kg (63.92 μg) in children undergoing elective anaesthesia and surgery [105]. Solid line, 50th percentile of simulated plasma concentrations; Shadow, 5th–95th interval of the simulated plasma concentrations.

**Figure 5 pharmaceutics-17-00214-f005:**
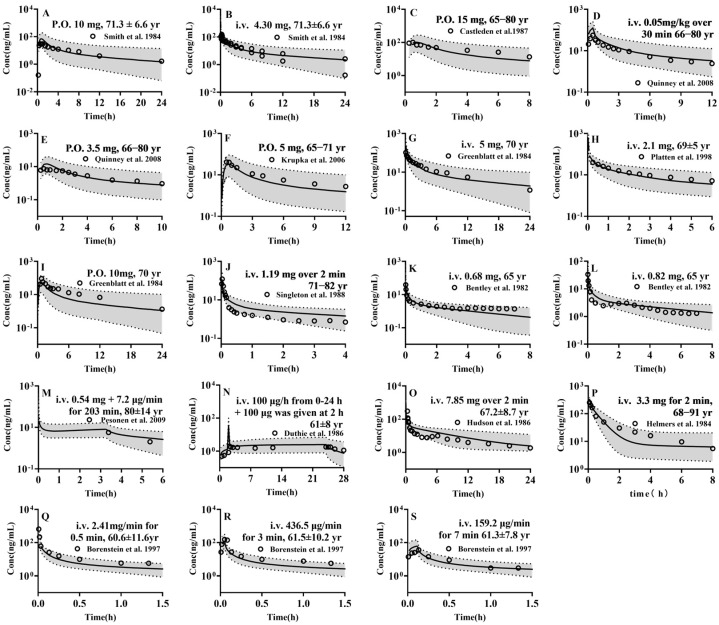
The predicted (lines) and observed (points) plasma concentrations of (**A**–**I**) midazolam, following oral administration of midazolam (**A**) 10 mg [100], (**C**) 15 mg [74], (**E**) 3.5 mg [106], (**F**) 5 mg [107], (**I**) 10 mg in healthy elderly [71] and intravenous administration (**B**) 0.07 mg/kg (4.30 mg) [108], (**D**) 0.05 mg/kg over 30 min [106], (**G**) 5 mg over 10 min [71], (**H**) 0.05 mg/kg in healthy elderly [109], (**J**–**O**) fentanyl, following intravenous administration of fentanyl (**J**) 15.5 μg/kg over 2 min in patient undergoing nonabdominal surgery [110], (**K**) [85] to (**L**) 10 μg/kg [85], (**M**) 7.5 μg/kg (0.54 mg) bolus continued with 0.1 μg/kg/min for 203 min in patient undergoing cardiopulmonary bypass surgery [111], (**N**) 100 μg/h over 24 h plus a bolus dose 100 μg at 2 h in patient undergoing orthopaedic surgery [112], (**O**) 100 μg/kg for 2 min in patient undergoing elective abdominal aortic surgery [113], (**P**) alfentanil, following intravenous administration of alfentanil 50 μg/kg over 2 min in patient undergoing intra-abdominal surgery [87], (**Q**–**S**) sufentanil, following intravenous administration of sufentanil (**Q**) 30 μg/kg/min for 0.5 min [114], (**R**) 5 μg/kg/min for 3 min [114] and (**S**) 2 μg/kg/min for 7 min in patient undergoing aortocoronary bypass surgery [114]. Solid line, 50th percentile of simulated plasma concentrations; Shadow, 5th–95th interval of the simulated plasma concentrations.

**Figure 6 pharmaceutics-17-00214-f006:**
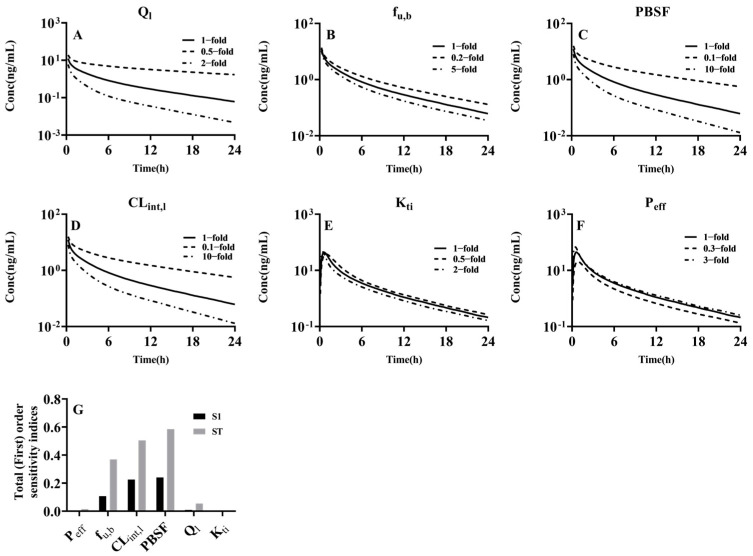
Sensitivity analysis parameters for midazolam. Local sensitivity analysis of variations in Q_l_ (**A**), f_u,b_ (**B**), PBSF (**C**) and CL_int,l_ (**D**) in plasma concentrations of midazolam following intravenous doses of 0.75 mg midazolam over 2 min to adults. A local sensitivity analysis of variations in K_ti_ (**E**) and P_eff_ (**F**) on plasma concentrations of midazolam following oral doses of 10 mg midazolam for adults. The solid black lines mark the lines of identity. Black dashed lines indicate the reduction factor, and black dotted lines indicate the magnifying factor. A global sensitivity analysis of P_eff_, f_u,b_, and CL_int,l,_ PBSF, Q_l_ and K_ti_ (**G**). Total order sensitivity indices (grey bar) and first-order sensitivity indices (black bar) were shown, respectively.

**Figure 7 pharmaceutics-17-00214-f007:**
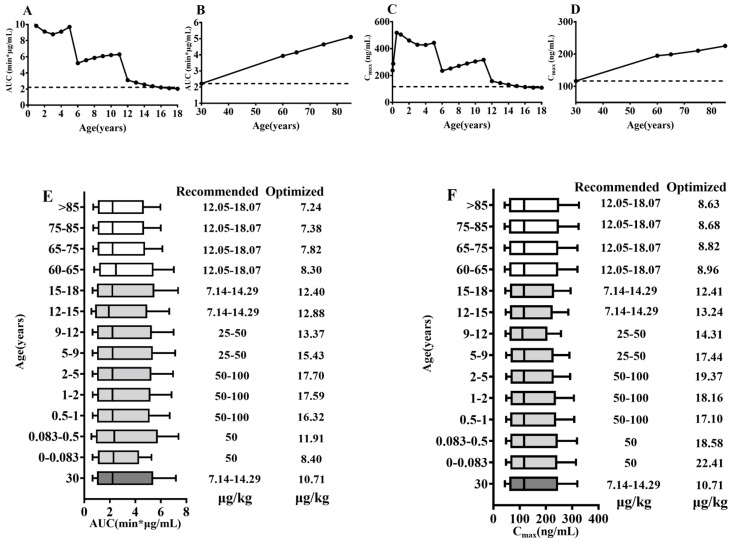
The AUC (**A**,**B**) and C_max_ (**C**,**D**) of midazolam at the clinically recommended administered doses. A Box–Whisker analysis of midazolam with an adjusted dose design based on AUC (**E**) and C_max_ (**F**). Black dashed lines indicate the parameter values of adults; solid black lines indicate the parameter values of pediatrics and the elderly.

## Data Availability

Data are contained within the article and the Appendix A.

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
