# Peer review of "Prediction of Pharmacokinetics for CYP3A4-Metabolized Drugs in Pediatrics and Geriatrics Using Dynamic Age-Dependent Physiologically Based Pharmacokinetic Models"

_pharmaceutics, 2025, doi:10.3390/pharmaceutics17020214_

Round 1

Reviewer 1 Report

Comments and Suggestions for Authors

This manuscript is titled “Prediction of pharmacokinetics for CYP3A4-metabolized drugs in pediatrics and geriatrics using the dynamic age-dependent physiologically based pharmacokinetic model” and presents valuable insight into predicting the pharmacokinetics of CYP3A4-metabolized drugs in different age groups and optimizing their dosage regimen in pediatrics and geriatrics. The research is both valuable and interesting. The author optimized a well-established PB-PK medal, which may be applied to pediatrics and geriatrics. The manuscript is accepted in its current form; however, here are some suggestions that could further enhance the scientific soundness and overall quality of the manuscript:

1.      The manuscript lacks a conclusion section. Please include and summarize the applications and implications of this study.

2.      Why did the author not discuss gender differences during model optimization? Could there be any sex-related differences observed in PK and PD for these model drugs? Please address this in the manuscript.

3.      Does the rate of CYP3A4 metabolism differ due to gender differences? Please discuss it in the manuscript.

4.      Why did the authors not include %CV (coefficient of variation) analysis for observed and predicted value comparisons?

5.      The discussion section should be revised to be more concise.

Comments on the Quality of English Language

NA

Reviewer 2 Report

Comments and Suggestions for Authors

The manuscript entitled "Prediction of pharmacokinetics for CYP3A4-metabolized drugs in pediatrics and geriatrics using dynamic age-dependent physiologically based pharmacokinetic model", submitted by Jing Han et al, contains an impressive amount of simulated (generated by "age-dependent PBPK model") data confronted with a consistent amount of literature reported experimental data. The immediate result is the validation of the proposed pharmacokinetic model for "normal" healthy patients, for which trial tests are available. The final aim consists in the model extrapolation to the pediatric and geriatric zones, where drug tests are scarce and much more difficult to perform. The obtained results are promising and point to the continuation of the research with model parameters refining on the pediatric side. 

The attached reviewed files contain some comments, suggestions and observations that may improve the manuscript.

Comments on the Quality of English Language

There are a few confusing formulations where rephrasing is needed. Some alternatives were proposed. 

A thorough check of the whole manuscript and supplementary file is recommended.

Reviewer 3 Report

Comments and Suggestions for Authors

The authors have developed a PBPK model for CYP3A4 substrates in adults to predict the exposure in paediatrics and geriatric patients, including age-dependent mechanism for relevant PK parameters. Overall, the strategy is acceptable, but additional comments are raised:

-The evaluation of model predictions is based on (i) The observed plasma concentration-profiles fell within the 5th-95th percentiles of simulations based on 1,000 virtual individuals; (ii) The simulations using the developed PBPK model were within the 0.5-2.0-fold of clinic observations. First, no inter-individual random effects were not evaluated nor justified with clinical evidence, just by allowing a variation between 0.67 to 1.5-fold of the ideal values at 120 the corresponding age-stages. If large inter-individual variability is included, every clinical data will fell within the prediction intervals, but it does not demonstrate the model is adequate. 

-1000 virtual patients are simulated at each step for evaluating the model, but mean data is used in most of the cases. Therefore, alternative metrics should be considered (PE or AFE) in order to evaluate whether mean/median predictions are able to predict the observed data. 

-Alternative ontogeny functions for ALB are available in the literature. Additional justification of the function selected should be provided.

-Did authors perform a global-sensitivity analysis for the identification of the more accurate parameters contribution?

Round 2

Reviewer 3 Report

Comments and Suggestions for Authors

The authors have addressed all the concerns